# Exogenous CFH Modulates Levels of Pro-Inflammatory Mediators to Prevent Oxidative Damage of Retinal Pigment Epithelial Cells with the At-Risk CFH Y402H Variant

**DOI:** 10.3390/antiox12081540

**Published:** 2023-07-31

**Authors:** Henry Velazquez-Soto, Sergio Groman-Lupa, Marisa Cruz-Aguilar, Alberto L. Salazar, Juan C. Zenteno, Maria C. Jimenez-Martinez

**Affiliations:** 1Department of Immunology, Research Unit, Institute of Ophthalmology “Conde de Valenciana Foundation”, Mexico City 06800, Mexico; henry.velazquez@institutodeoftalmologia.org (H.V.-S.);; 2Department of Genetics, Institute of Ophthalmology “Conde de Valenciana Foundation”, Mexico City 06800, Mexico; 3Department of Biochemistry, Faculty of Medicine, National Autonomous University of Mexico, Mexico City 04510, Mexico

**Keywords:** CFH, RPE, oxidative stress, AMD, cytokines, cell death

## Abstract

Age-related macular degeneration (AMD) is a complex, progressive degenerative retinal disease. Retinal pigment epithelial (RPE) cells play an important role in the immune defense of the eye and their dysfunction leads to the progressive irreversible degeneration of photoreceptors. Genetic factors, chronic inflammation, and oxidative stress have been implicated in AMD pathogenesis. Oxidative stress causes RPE injury, resulting in a chronic inflammatory response and cell death. The Y402H polymorphism in the complement factor H (CFH) protein is an important risk factor for AMD. However, the functional significance of CFH Y402H polymorphism remains unclear. In the present study, we investigated the role of CFH in the pro-inflammatory response using an in vitro model of oxidative stress in the RPE with the at-risk CFH Y402H variant. ARPE-19 cells with the at-risk CFH Y402H variant were highly susceptible to damage caused by oxidative stress, with increased levels of inflammatory mediators and pro-apoptotic factors that lead to cell death. Pretreatment of the ARPE-19 cell cultures with exogenous CFH prior to the induction of oxidative stress prevented damage and cell death. This protective effect may be related to the negative regulation of pro-inflammatory cytokines. CFH contributes to cell homeostasis and is required to modulate the pro-inflammatory cytokine response under oxidative stress in the ARPE-19 cells with the at-risk CFH Y402H variant.

## 1. Introduction

Age-related macular degeneration (AMD) is a leading cause of legal blindness worldwide and is predicted to affect 288 million people by 2040 [1,2]. AMD affects the macula region of the retina, causing the progressive loss of central vision [3]. Early stages of AMD are characterized by drusen and other alterations in the retinal pigment epithelium; meanwhile, late stages may lead to neovascular or non-neovascular forms of the disease, leading to a severe and irreversible loss of vision [4]. Several factors, such as genetic predisposition, chronic local inflammation, and oxidative stress, contribute to the etiology of AMD; however, the mechanisms leading to AMD development are not fully understood [5,6,7].

Retinal pigment epithelium (RPE) cells are crucial for the maintenance of retinal homeostasis because they participate in several key functions, such as the transport of nutrients, preservation of the retinal structure, phagocytosis of the outer segments of photoreceptors, and protection of the retina from oxidative damage [8,9]. The RPE is surrounded by a highly oxidized microenvironment generated by a combination of visible light exposure, elevated metabolic activity, and an accumulation of oxidized lipoproteins [10,11]. Physiological functions and aging processes, in combination with environmental stressors, including smoking or obesity-related conditions, force RPE cells to support excessive levels of oxidative stress, disturbing cell homeostasis [12] and causing chronic inflammation [13].

Genetic predispositions play an essential role in AMD pathogenesis. Different variants of complement system proteins are associated with an increased risk of AMD development and progression [14]. Withing a near 40% prevalence [15], the single nucleotide polymorphism predicting a Y402H replacement in the gene coding complement factor H (CFH) is an important risk factor for AMD [16]. CFH is a primary inhibitor of complement alternative pathways. By acting as a cofactor for factor I, it prevents the formation of C3 convertase through binding to C3b [17,18]. CFH is present in human and mouse ocular tissues, such as the RPE and choroid. Moreover, CFH and other complement proteins have been found in the drusen of patients with AMD, suggesting that inflammation is an important component of this disease [19].

Although the precise mechanism through which the CFH Y402H variant confers an increased risk of AMD remains unclear, in vitro studies have shown that the at-risk CFH Y402H variant leads to the dysregulation of the complement pathway, with increased levels of the C3 protein [16]. Interestingly, the Y402H polymorphism is located at short consensus repeat (SCR) 7 of the protein, which is important for mediating CFH binding to polyanions, such as heparan sulfate chains, C-reactive proteins, malondialdehyde, and glycosaminoglycans. Moreover, inefficient binding of the at-risk CFH Y402H variant to malondialdehyde induces macrophages and monocytes to secrete many pro-inflammatory cytokines, suggesting that systemic cytokines may also be associated with genetic factors and, thus, contribute to AMD [20,21]. This risk variant is associated with high levels of complement activation products and proinflammatory cytokines in the blood of patients with AMD [22].

Similarly, studies on the RPE in vitro showed that the poorly binding affinity of CFH Y402H to C-reactive protein resulted in elevated complement activation and altered the secretion of the pro-inflammatory cytokines, including tumor necrosis factor-alpha (TNF-α) [20,21], and led to the upregulation of oxidative stress markers [22,23]. Several animals and in vitro models have been used to investigate the association between CFH and AMD. Mouse models provide a conduit for dissecting the molecular mechanisms linking complement responses to AMD pathobiology [24]. Given the complexity of the disease, none of the available models fully recapitulate human AMD; however, they have provided important insights into the roles of the complement system and CFH in AMD. The in vitro generation of RPE monolayers from primary donor eye cultures has made it possible to study cell dynamics under experimental conditions, which has increased our knowledge of the cellular and molecular mechanisms of AMD. However, the limited life span and absence of the at-risk CFH variant in normal human RPE cells represent an obstacle to molecular analysis.

Commercially available, spontaneously generated human RPE cell line (ARPE-19) cells are a valuable source of human RPE cells. As the ARPE-19 cell line has been shown to maintain many of the properties of the native cells of the RPE, it has frequently been used as an in vitro model for the study of retinal disorders, such as AMD. Previous assays using ARPE-19 cells showed that an oxidized environment increases the expression of complement receptors and accumulated C3, CFH, properdin, and NLRP3, act as a mechanism to maintain cell homeostasis following oxidative stress [25].

With all of these backgrounds raised, we explored the role of CFH in regulating cell death and inflammation induced by oxidative stress conditions in RPE cell cultures with the at-risk CFH Y402H variant.

## 2. Materials and Methods

### 2.1. Cell Culture

The ARPE-19 was purchased from the American Type Culture Collection (ATCC CRL-2302, Manassas, VA, USA) and cultured in 25 mL flasks containing Dulbecco’s modified Eagle’s nutrient F12 medium (DMEM/F12, STEMCELL Technologies) supplemented with 10% fetal bovine serum (GIBCO, BRL), 100 U/mL penicillin, 100 mg/mL streptomycin (GIBCO, BRL), 1mM glutamate (Stem cell, Sartorius), and sodium pyruvate (Stem cell, Sartorius). The cell culture was kept under a humid atmosphere of 37 °C and 5% carbon dioxide (CO_2_). Cells were harvested and adjusted to 2.5 × 10^5^ cells; then, they were seeded in 24-well plates until they reached 80% confluence.

### 2.2. Genomic DNA Extraction (gDNA)

The gDNA was obtained from 5 × 10^6^ cells using a Blood and Tissue Extraction Kit (QIAGEN, Hilden, Germany). The cells cultured in 25 mL culture flasks were dissociated with trypsin/EDTA, as previously described, and the cell button was resuspended in 200 μL of 1X PBS to continue with the manufacturer’s protocol. The concentration and purity of the extracted gDNA were determined by spectrophotometry (Nanodrop, Thermofisher, Waltham, MA, USA); its integrity was verified by electrophoresis in 1.5% agarose gel, stained with SYBR^®^ Gold Nucleic Acid Gel Stain (Thermofisher Scientific) for visualization under UV light in a photo documenter.

### 2.3. ARPE-19 Genotyping

The region of the CFH gene that contains the single nucleotide polymorphism (SNP) rs1061170 (c.1204T>C; p.Tyr402His) was amplified by a polymerase chain reaction (PCR) using the oligonucleotides CFH_Fw:5′-AGT TCG TCT TCA GTT ATA C-3′ and CFH_Rv:5′-TGG TCT GCG CTT TTG GAA GAG c-3′. The amplification reaction consisted of a mixture of 60 ng of gDNA, 5 μL of buffer A2 KAPA2G 5X, 0.5 μL of KAPA dNTP Mix 10 mM, 1.25 μL of each 10 μM oligonucleotide, 0.1 μL of KAPA2G Fast DNA Polymerase 5 U/μL, and sufficient nuclease-free water to achieve a 25 μL reaction. The amplification conditions consisted of an initial denaturation cycle at 95 °C for 3 min, followed by 38 cycles consisting of denaturation at 95 °C for 1 min each, hybridization at 59.5 °C for 30 s, extension at 72 °C for 1 min, and a final extension cycle at 72 °C for 3 min. Five microliters of the PCR product was analyzed on a 1.5% agarose gel and was electrophoresed. The remaining samples were purified using a DNA Clean and Concentrator Kit (ZYMO Research, Irvine, CA, USA). Genotyping of the CFH variant was carried out by automated direct sequencing with the BigDye Terminator Cycle Sequencing Kit (Applied Biosystems, Waltham, MA, USA), following the manufacturer’s protocol and using a temperature program consisting of 25 denaturation cycles at 97 °C for 30 s, alignment at 50 °C for 30 s, and an extension at 60 °C for 1 min. The samples were analyzed using an ABI Prism 3130 Genetic Analyzer Sequencer (Applied Biosystems) and the obtained sequences were compared with a reference published in ENSEMBL (NM_000186.4).

### 2.4. RNA Extraction and cDNA Synthesis

Total RNA was extracted from 1 × 105 ARPE-19 cells using Trizol assay [25]. These samples were treated with RNase-free DNase I (QIAGEN), according to the manufacturer’s protocol. cDNA was synthesized from 2 μg of total RNA in a final volume of 20 μL using the Superscript III First-Strand Synthesis System (Thermo Fisher Scientific) with oligo (dT). A PCR was carried out with 125 ng of cDNA as a template in a 15 μL reaction mix containing Kapa Taq Readymix PCR Kit (Sigma-Aldrich, St. Louis, MO, USA) and the specific primers for CRALBP (CRALBP_Fw:5′-TGGTGTCCTCTCTAGTCGGG-3′, CRALBP_Rv:5′-GTTCAGCTGGCAGGAGATGT-3′) and RPE65 (RPE65_Fw:5′-ACAACTTGGCCCTGACTTCC-3′, RPE65_Rv:5′-TGAAGAGCATGACCACTCGG-3′). The transcripts were analyzed using the GeneAmp PCR System 9700 Thermocycler (Applied Biosystem) and PCR conditions of 35 cycles formed by denaturation at 95 °C for 30 s, hybridization at 60 °C for 15 s, and extension at 72 °C for 1 min. Furthermore, the amplified products were analyzed by electrophoresis in agarose gel.

### 2.5. Flow Cytometry

The cultured ARPE-19 cells were analyzed using flow cytometry. The cells were harvested by treatment with 0.25% trypsin- 0.5% EDTA (Gibco, Carlsbad, CA, USA), washed with 1X PBS, and the cell pellet was resuspended in 2% fetal bovine serum in 1X PBS for 1 h at room temperature.

The identification of surface molecules was carried out by incubating the cells with human monoclonal APC-conjugated anti-toll like receptor 4 (TLR4), FITC-conjugated CD14, and PE-conjugated CD86 antibodies (all from Biolegend, San Diego, CA, USA) for 20 min at 4 °C, according to the manufacturer’s recommendations. Cells were washed with 1X PBS and fixed with 4% paraformaldehyde (Sigma-Aldrich, Inc.) for 20 min at room temperature, washed, and acquired with a BD FACSVerse Flow Cytometer (BD Biosciences, San Jose, CA, USA). The results were analyzed using Flow Jo ver. 10 (Beckton Dickinson, East Rutherford, NJ, USA).

### 2.6. Induction of Oxidative Stress and Cell Viability Assay

To induce an oxidative stress condition, the cells were treated with 400 μM or 800 μM of hydrogen peroxide (H_2_O_2_) for 24 h. The concentrations for H_2_O_2_ were selected according to previous reports [26,27,28]. Thereafter, the cells were harvested, washed, and resuspended in 1X phosphate-buffered saline (PBS) for treatment and subsequent analysis.

The effect of oxidative stress on ARPE-19 cell viability was evaluated by staining them with 7-amino-actinomicin (7-AAD; Molecular Probes, Invitrogen, Carlsbad, CA, USA). Briefly, living cells keep their membranes intact, which prevents the passage of the 7-AAD dye; in non-viable cells, the membranes are damaged and permeable. Therefore, the dye easily penetrates and binds to double-stranded DNA. ARPE-19 cells were cultured with or without the addition of 1, 2 or 5 µg/mL of exogenous CFH (Sigma Aldrich, San Louis, MO, USA) and in the presence or absence of 400 μM of H_2_O_2_ as an inducer of oxidative stress. Evaluated concentrations for exogenous CFH ranged around the previously reported concentration of 3 µg/mg found in intravitreal human samples [29]. In some experiments, CFH was added to the cell culture 1 h before or simultaneously with H_2_O_2_. Cells with or without stimuli were harvested and washed twice in 1X PBS containing 2% BSA and were subsequently resuspended in 100 μL of 1X PBS. The exclusion of viable cells was carried out by staining with 5 μL of 7-AAD for 5 min, according to the manufacturer’s instructions; this was subsequently analyzed by flow cytometry. Unstained cells were included in each experiment as an autofluorescence reference control.

### 2.7. Intracellular for Staining Cytochrome C and NF-κB

Nuclear factor kappa b (NF-κB) and the release of cytochrome C from the intermembrane space of the mitochondria into the cytosol was measured using a monoclonal fluorescein isothiocyanate (FITC)-conjugated anti-NF-κB or FITC-conjugated anti-cytochrome C antibody (Biolegend, San Diego, CA, USA). ARPE-19 cells treated with H_2_O_2_ for 3, 6, 12, or 24 h were permeabilized and subsequently fixed with 1% PFA in 1X PBS. The mean fluorescence intensity (MFI) for both molecules was measured by flow cytometry and analyzed using FlowJo software, ver.10 (Beckton Dickinson).

### 2.8. Cytokine Measurement

The cytokines produced after the induction of oxidative stress in ARPE-19 cells, with or without the addition of exogenous human plasma complement factor H (CFH; Sigma Aldrich, Saint Louis, MO, USA) at different concentrations (1, 2.5, and 5 µg/mL), were determined using the cytometric bead array (CBA) Human Inflammatory Cytokines Kit (BD Biosciences), according to the manufacturer’s specifications. The assay allowed the simultaneous and quantitative measurement of cytokines, such as interleukin (IL)-8, IL-1β, IL-6, IL-10, TNF-α, and IL-12p70, in the cell culture supernatant. The induced culture supernatants were added to a mixture of capture antibodies, bead reagents, and PE-conjugated detection antibodies. The mixture was then incubated at room temperature, in the dark and was washed. The cytokine panel was analyzed using FCAP Array Software v.3. (BD Biosciences). Detection limits for this method were as follows: 3.6 pg/mL of IL-8, 7.2 pg/mL of IL-1β, 2.5 pg/mL of IL-6, 3.3 pg/mL of IL-10, 3.7 pg/mL of TNF-α, and 1.9 pg/mL of IL-12p70.

### 2.9. Statistics

All experiments were performed independently, at least three times, under each condition. All analyses were performed using GraphPad Prism 5.0 (GraphPad Software Inc., La Jolla, CA, USA). A normality test was performed on all of the data obtained from the previous ANOVA and post hoc statistical analyses. Statistical significance was set at a *p*-value of less than 0.05.

## 3. Results

### 3.1. ARPE-19 Cell Genotyping

Although ARPE-19 cells are commercially available and widely used for research, there are no previous studies where this cell line has been proven to carry the CFH Y402H variant.

Thus, we initially determined the presence of the genetically at-risk CFH Y402H variant in ARPE-19 cells by direct sequencing. The gDNA of ARPE-19 cells was used as a template to amplify exon 9 of CFH by PCR. After the sequencing the PCR product, we identified a heterozygous missense variant consisting of a c.1204 T to C transition, which was predicted to change tyrosine (TAT) to histidine (CAT) at residue 402 (Figure 1). Sequencing for CFH of variant carrying individual and wildtype genotype are shown in Appendix A.

We characterized the ARPE-19 cell line with respect to the expression of differentiation markers, such as retinaldehyde-binding protein (CRALBP) and protein-specific RPE (RPE65), by a real-time polymerase chain reaction (RT-PCR). After the amplification of the transcripts for differentiation markers, they were separated by electrophoresis into single transcripts of 593 bp for CRALBP and 598 bp for RPE65 (Figure 2).

The presence of this genetic variant that confers the risk of developing AMD and the expression of differentiation markers validated the idea that the ARPE-19 cell line could be suitable for establishing an in vitro model of oxidative stress and studying the participation of the Y402H genetic variant in the generation of an inflammatory environment.

### 3.2. Identification of Molecules of Immunological Importance in ARPE-19

The expression of molecules of potential immunological importance on the surface of ARPE-19 cells was determined by detection, with specific antibodies conjugated to different fluorochromes, and was analyzed by flow cytometry.

Under stable conditions and in the absence of external stimuli, ARPE-19 cells constitutively express basal levels of receptors, such as TLR4 and its co-receptor CD14. While CD86 showed low basal expression levels, TLR4 was the most highly expressed molecule, followed by CD14. Representative histograms for the detection of the molecules analyzed in this study are shown in Figure 3.

### 3.3. H_2_O_2_-Induced Cell Death

ARPE-19 cells without H_2_O_2_ treatment were used as controls, representing more than 80% of viable cells. In contrast, the treatment of cells with H_2_O_2_ significantly decreased the membrane permeability of more than 70% of the cells (Figure 4).

The representative dot plots in Figure 4A,B show the distribution of ARPE-19 cells before and after the induction of oxidative damage and subsequent treatment with 7-AAD. All samples were analyzed using the same gates. Within the stressed cells, living cells (30%) and apoptotic cells (70%) were identified. Our results indicate that the H_2_O_2_ concentration used in this assay induced an oxidative stress state in ARPE-19 cells, which significantly affected membrane permeability and compromised cell viability (Figure 4C).

### 3.4. Release of Apoptotic and Inflammatory Markers Induced by Oxidative Stress in ARPE-19 Cells

Intracellular molecules were identified using specific antibodies and analyzed using flow cytometry. The mean fluorescence intensity (MFI) was determined at different times of oxidative stress induction (Figure 5).

Figure 5A,B show an increase in the level of cytochrome C released by ARPE-19 cells after exposure to H_2_O_2_. The release of cytochrome C into the cytosol increases in a time-dependent manner. In the early stages of oxidative stress induction, the cytosolic cytochrome C content was low and increased over time, reaching a maximum at 24 h. The cytosolic content of cytochrome C at 24 h post-oxidative stress was four times higher post-treatment. Furthermore, the induction of NF-κB, as shown in Figure 5C,D, correlated with the levels of cytoplasmic cytochrome C. The maximum synthesis of these components was observed 24 h after subjecting the ARPE-19 cells to oxidative stress.

Taken together, these results suggest that ARPE-19 cells harboring the at-risk CFH Y402H variant are highly susceptible to oxidative-stress-induced damage. In response to oxidative damage, pro-inflammatory and pro-apoptotic factors, such as cytochrome C and NF-κB, can be detected.

### 3.5. Exogenous CFH Protein Protects ARPE-19 Cells from Damage Caused by Oxidative Stress

As we observed that oxidative stress affected the ARPE-19 cells’ viability, we considered it important to evaluate whether the presence of exogenous CFH could rescue cells from oxidative damage and prevent apoptosis.

Thus, to evaluate the effect of CFH on cell viability after the induction of oxidative stress, ARPE-19 cells were cultured in the presence of any of the following treatments: a simultaneous combination of exogenous CFH and H_2_O_2_ for 24 h or a treatment for 1 h with exogenous CFH, followed by H_2_O_2_ induced-oxidative stress for 24 h. Cell viability was evaluated by flow cytometry to determine the percentage of cells that were negative for 7-AAD staining. The data were then compared with those obtained from cells cultured in the presence of exogenous CFH or H_2_O_2_ as the only stimuli. ARPE-19 cells without any treatment were used as positive controls.

Figure 6 shows the percentage of viable cells after 24 h of treatment. ARPE-19 cells treated with exogenous CFH did not show deleterious changes and maintained a level of viability similar to that of unstimulated cells (80%). In contrast, the induction of oxidative stress by H_2_O_2_ significantly affected cell survival (20% of viable cells), as shown in previous studies.

The addition of exogenous CFH to the cell culture increased the percentage of viable cells when added simultaneously with, or before, the induction of oxidative stress. The simultaneous addition of exogenous CFH and H_2_O_2_ to the cell culture increased the percentage viability (55%) almost two-fold, with respect to the condition where only oxidative stress was induced. Similarly, when the cultures were treated with exogenous CFH and subsequently subjected to oxidative stress, the percentage of viable cells (74%) increased 2.3 times, with respect to cells treated only with H_2_O_2_, or 1.3 times, with respect to cells treated simultaneously with exogenous CFH and H_2_O_2_.

However, when exogenous CFH was incorporated into the culture medium, ARPE-19 cells could modulate the effect of oxidative stress. This cellular protection was more effective when CFH was present in the cell culture before stimulation with oxidative stress. Furthermore, although exogenous CFH contributed to the protection of cells against oxidative damage, the percentage of viable cells did not reach the level observed in untreated cells.

### 3.6. Effect of Exogenous CFH on the Synthesis of Pro-Inflammatory Cytokines Induced by Oxidative Stress

ARPE-19 cells were treated for 1 h with exogenous CFH (5 µg/mL) prior to the addition of H_2_O_2_, or (1, 2.5, and 5 µg/mL) simultaneous to the addition of H_2_O_2_, maintaining the stress condition for 24 h. Additionally, we evaluated the cytokine profile depending on the concentration of exogenous CFH added simultaneously with H_2_O_2_ to the cell culture. Cytokines in the supernatants of the ARPE-19 cell cultures were quantified using the CBA method (Figure 7). It is important to note that under some conditions, the evaluated cytokines showed concentrations below the estimated detection limit.

Oxidative stress did not induce significant changes in the synthesis of IL-1β and IL-12. Even though the synthesis of these showed an increased expression trend with respect to unstimulated cells, it was not statistically significant.

The induction of oxidative stress triggered the significant overexpression of IL-8, which increased 30 times with respect to non-stimulated cells or those stimulated with exogenous CFH. This shows that the synthesis of IL-8 is inhibited by CFH, regardless of the concentration. Similarly, exogenous CFH regulates IL-8 synthesis when present in the cell culture before or simultaneously with H_2_O_2_. Furthermore, IL-6 also significantly increased after the induction of oxidative stress compared with when 5 μg of exogenous CFH was added previously or simultaneously with H_2_O_2_.

The synthesis of TNF-α was increased 2.3 times in ARPE-19 cells under oxidative stress. The treatment of cells with exogenous CFH at the highest concentration tested, added before or simultaneously with H_2_O_2_, significantly decreased (approximately three times) the synthesis of TNF-α with respect to the treatment of cells with only H_2_O_2_.

Finally, IL-10, an anti-inflammatory cytokine, was synthesized in response to oxidative stress similar to levels in untreated cells. Lower IL-10 concentrations were observed when exogenous CFH was added in combination with H_2_O_2_. The addition of exogenous CFH to cell cultures inhibited the IL-10 synthesis induced by oxidative damage.

These results indicate that oxidative stress being induced in ARPE-19 cells causes the expression of pro-inflammatory cytokines, which are regulated by exogenous CFH, suggesting that CFH is important for maintaining cell homeostatic function (Figure 8).

## 4. Discussion

The RPE performs different complex functions that are necessary to maintain photoreceptor homeostasis. Alterations in the RPE cell layer lead to deleterious effects and photoreceptor death, triggering diseases, such as AMD. The risk of developing AMD is mainly due to advanced age, environmental stress, and genetic factors [7,23,30]. Among genetic factors, the at-risk CFH Y402H variant is strongly associated with AMD [31,32,33]. However, little is known about the impact of this variant on RPE cell death and inflammation, particularly cytokine production regulation.

The main interest of the present work aimed to study the role of CFH and its ability to regulate RPE cell death and proinflammatory cytokines following oxidative stress, as is proposed to occur in the AMD microenvironment. Therefore, after selecting the ARPE19 cell line for our experiments, the commercially available and most-used RPE cell line, we decided to genotype the region of the CFH gene that contains the single nucleotide polymorphism (SNP) rs1061170 (c.1204T>C; p. Tyr402His), a variant previously reported in multiple AMD patients’ cohorts [6,14,33]. Interestingly we found and reported, for the very first time, that ARPE-19 cells carry the at-risk variant Y402H, making this cell line even more suitable for the research of complement role in AMD, particularly for understanding CFH involvement.

The ARPE-19 cell line used in this study maintained many of the differentiation characteristics of RPE, such as the expression of RPE65 and CRALBP, which are involved in the vitamin A cycle and regeneration of visual pigment, respectively [34]. Molecules, such as TLR4 and its CD14 co-receptor, were identified as part of their cellular characterization. It is well known that the toll-like receptors and co-stimulatory molecules expressed in RPE cells are important for inducing ocular inflammation mechanisms because of their ability to secrete or potentiate various inflammatory mediators [35,36,37]. The heterozygous presence of the Y402H variant allowed us to consider them an ideal cell model for studying the effect of the at-risk CFH variant on the inflammatory response to oxidative stress.

H_2_O_2_ is known to be one of the primary causes of oxidative stress in RPE cells. In this study, we used H_2_O_2_ to mimic the conditions of physiological oxidative stress in vitro and evaluate the role of the at-risk CFH Y402H variant in ARPE-19 cells in generating the inflammatory cytokine profile under these conditions. The RPE is susceptible to oxidative stress but is protected by the production of antioxidant agents and its ability to regulate apoptosis under certain conditions [10]. However, the inadequate neutralization of oxidative stress affects mitochondrial function and can lead to cellular damage. The main event after damage to the mitochondria is increased permeability and loss of membrane potential, which allows the release of pro-apoptotic molecules, such as cytochrome C, into the cytoplasm [38,39,40]. Furthermore, the activation of caspases 9 and 3 has been demonstrated in RPE cells treated with H_2_O_2_ [39]. In our oxidative stress model using H_2_O_2_, ARPE-19 cells showed different degrees of alteration in the permeability of their plasma membrane, which correlated with the amount of the internalized fluorescent marker (7-AAD) and allowed for the identification of living cells with intact plasma membranes and apoptotic cells. The increased release of cytochrome C in stressed cells confirmed apoptosis. This release increased in a time-dependent manner, which correlated with the permeability of the plasma membrane.

Intervention with caspases and the proteolytic pathway of the Bcl-2 family could be involved in the cytochrome C release. This could be feasible considering that in the mitochondria, Bcl-2 proteins modulate the apoptotic signal that releases apoptotic factors, such as cytochrome C, and activates effectors, such as caspases 9 and 3, that promote cell death [41,42]. Additionally, signal transduction through Fas or TNF-α receptors with caspase 8 activation also stimulates cytochrome C release through the cleavage pathway of Bcl-2 family products [43]. Our assays showed that the release of cytochrome C was accompanied by the expression of the NF-κB transcription factor. These results suggest that oxidative stress triggers a signal transduction cascade that induces the expression of genes encoding NF-κB-dependent pro-inflammatory cytokines. However, further studies are needed to investigate the possible signaling pathways that lead to apoptosis.

Oxidative stress and the inflammatory process are known to be associated with AMD [13,22]. When a cell damage signal is generated, caspase activation mechanisms are triggered, which subsequently activate the synthesis of pro-inflammatory cytokines. Since our previous results showed that the addition of exogenous CFH to the cell culture prevented apoptosis, induced by exposure to oxidative stress, we evaluated whether this protection was driven by the modulation of pro-inflammatory cytokines.

The synthesis of pro-inflammatory cytokines is dependent on the activation of transcription factors or the action of other stimuli leading to the establishment of a state of inflammation that contributes to the elimination of agents that cause cell damage; however, under disease conditions, exacerbation of the inflammatory response leads to cell and tissue damage. In RPE cells, the expression of IL-6 and IL-8, for example, is induced in vitro by IL-1b, a potent initiator of the immune response [44]. The synthesis of these cytokines by RPE cells plays an important role in the immune processes of the eye to achieve homeostasis in stressful situations; however, its exacerbated or poorly controlled action induces severe tissue damage, leading to the development of diseases, such as AMD [41].

Our results showed that ARPE-19 cells were highly susceptible to damage caused by oxidative stress, with increased levels of inflammatory mediators and pro-apoptotic factors that lead to cell death. However, it is unknown whether the presence of Y402H polymorphism is related to the inability of cells to achieve homeostasis under oxidative stress. Therefore, we evaluated whether the presence of exogenous wild-type CFH could restore the ability of cells to modulate inflammatory responses and survive under prolonged conditions of oxidative stress.

Exogenous CFH incorporated into the cell culture prior to the induction of oxidative stress prevented damage and cell death. This protective effect was related to the negative regulation of pro-inflammatory cytokines induced by exogenous CFH. In the absence of exogenous CFH, the synthesis of pro- and anti-inflammatory cytokines significantly increased, causing deleterious effects on cell viability. Similar alterations in cytokine profiles were observed when ARPE-19 cells were treated simultaneously in with both stimuli. However, it is important to note that, in some cases, the levels of cytokines were dependent on the concentration of exogenous CFH, such as IL-1β or TNF-α; high concentrations of exogenous CFH inhibited the synthesis of these cytokines. This differential behavior for cytokine secretion and regulation after adding exogenous CFH may be explained by transcriptional, post-transcriptional, and gene expression mechanisms [45], as has been reported in other RPE cell lines exposed to oxidative conditions [46].

In basal conditions, RPE cells express very low levels of IL-1β, which can be slightly induced by treatment with TNF-α or the antibody cross-linking of CD48 on the surface of cells, suggesting a minor role in the inflammatory process of the posterior segment of the eye [47]. However, in our study, no significant differences were found in the levels of IL-1β synthesis in the ARPE-19 cell supernatant after any treatments. These results may indicate that the availability of CFH in the environment before an oxidative insult occurs is required to modulate the pro-inflammatory cytokine response and protect cells from oxidative damage.

Therefore, it is possible that the elevation of pro-inflammatory cytokines under the oxidative stress observed in this study was related to the inability of the at-risk Y402H variant of CFH to regulate complement activation. Although our assays did not elucidate the mechanisms by which the at-risk variant in CFH promotes the dysregulated expression of pro-inflammatory cytokines, a plethora of evidence in the literature has been reported. Oxidative stress triggers the inflammatory response and activates death signals. The activation of complement and their bioactive fragments is also a potent stimulator of cytokine secretion that leads to the amplification of inflammatory responses [48].

Earlier studies have shown that the CFH Y402H polymorphism may be associated with increased activation of the complement cascade, where activation of complement proteins, such as C3a and C5a, may regulate the expression of pro-inflammatory cytokines [49,50]. Studies on ARPE-19 cells have shown increased intracellular concentrations of the complement regulators CFH, properdin, and central complement protein C3 following oxidative stress induction. Properdin serves as a positive complement regulator; therefore, a higher concentration of this protein in ARPE-19 cells results in enhanced cellular C3 cleavage [24]. It is potentially feasible that properdin activity prevails over the negative complement regulator CFH, which can be functionally decreased by the presence of the risk variant Y402H.

Other studies have reported that the complement activation product C5a is associated with the at-risk variant in the CFH gene and promotes the activation of the NF-κB pathway in the RPE in vitro [51]. Concurrently, the NF-κB pathway mediates the expression of classic inflammatory cytokines (IL-6, IL-8, TNF-α) and those dependent on the activation of the NLRP3 inflammasomes, such as IL-1β and IL-18 [46]. The addition of H_2_O_2_ to ARPE-19 cells increases the inflammasome activation NLRP3 and subsequently enhances the secretion of proinflammatory cytokines [25]. Thus, similar to our findings in ARPE-cells, it is possible that the CFH Y402H variant contributes to AMD via the enhanced activation of complement and NF-kB pathways, resulting in an increased secretion of inflammatory cytokines.

Some limitations of our work must be mentioned and considered to ease the interpretation, grasp the robustness of the work, and contribute to future research. The authors are aware that the main limitation in this report is the need for commercial RPE cells without the Y402H variant to act as a control. In addition, additional experiments are required to evaluate relevant complement molecules, such as C3, C5, and MAC formation (as previously proposed) [52], and the determination of endogenous CFH could help us to more precisely understand the mechanisms underlying the findings reported. Regarding cytokines, they ranged around the detection limits under certain experimental conditions and were not detected. Further studies are needed to evaluate, through distinct methodologies, cytokine involvement.

Despite the limitations described, the H_2_O_2_-induced oxidative stress model increases our knowledge of the cellular and molecular mechanisms involved in AMD related to the CFH Y402H variant and could be helpful in evaluating promising therapeutics, as reported in [53,54]. Additional research is required to verify the signaling pathway(s) induced by exogenous CFH to regulate the secretion of pro-inflammatory cytokines and to investigate the mRNA and protein levels of endogenous CFH expression. RNA related to complement and coagulation pathways, antigen presentation, tissue remodeling, and signaling pathways, such as toll-like, NOD-like, and PI3K-Akt, were reported in postmortem samples of AMD retinas by bulk transcriptomics [55]. In other RPE cell lines, mitochondrial RNA analysis has demonstrated the relevance of oxidative-induced apoptosis mechanisms related to ATP biosynthesis damage [56]. These data support the relevance of our findings.

## 5. Conclusions

This study demonstrates, for the first time, the presence of the CFH Y402H variant in the ARPE-19 cell line. We show that RPE cell cultures stimulated with H_2_O_2_ provide a valuable model for studying the conditions reflecting oxidative stress states; this allowed us to analyze phenomena, such as cell viability and the release of proapoptotic marker cytochrome c, the expression of NF-κB, and proinflammatory cytokine production.

Finally, our results show that CFH is essential for modulating the synthesis of pro-inflammatory cytokines and preventing cell damage in response to oxidative stress.

## Figures and Tables

**Figure 1 antioxidants-12-01540-f001:**
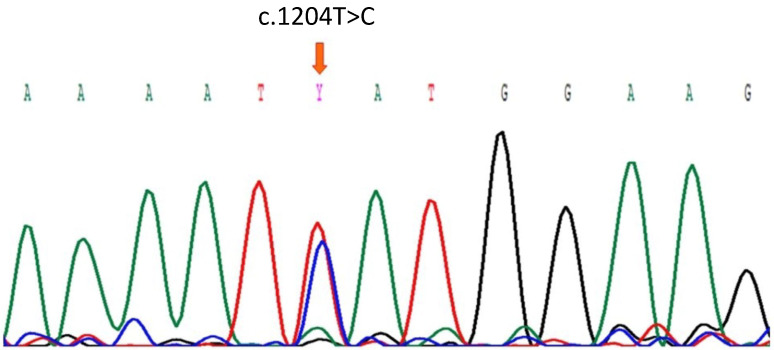
Partial Sanger sequence of the *CFH* gen showing the heterozygous c.1204T>C variant (arrow) in ARPE-19 cells.

**Figure 2 antioxidants-12-01540-f002:**
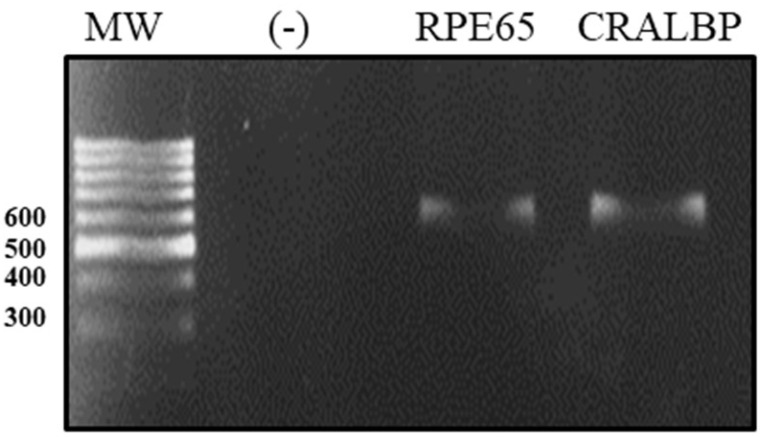
Electrophoretic run of ARPE-19 cell differentiation markers. The total RNA of ARPE-19 was extracted and the expressions of RPE65 and CRALBP were determined by RT-PCR. The CRALBP expression in peripheral blood monocytes was used as a negative control (−).

**Figure 3 antioxidants-12-01540-f003:**
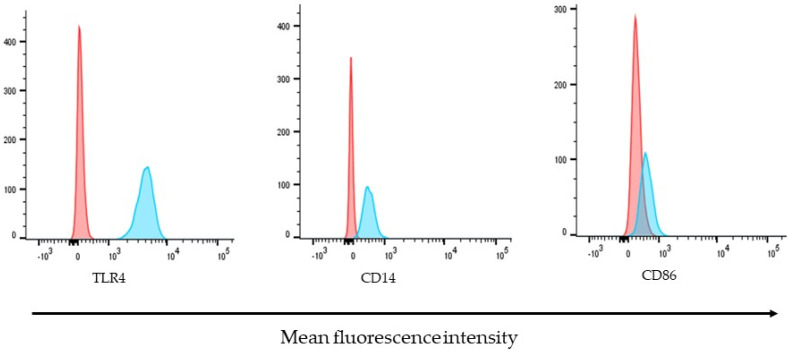
Molecules of immunological importance on the surface of ARPE-19 cells. The expression of TLR4, CD14, and CD86 were determined by their detection, with specific antibodies coupled to different fluorochromes. Histograms are representative of three independent assays and show the relative fluorescence intensity of ARPE-19 cells, unstained control (red) or stained (blue) with APC-conjugated-Anti-TLR4, FITC-conjugated-anti-CD40, or PE-conjugated-CD86 antibodies.

**Figure 4 antioxidants-12-01540-f004:**
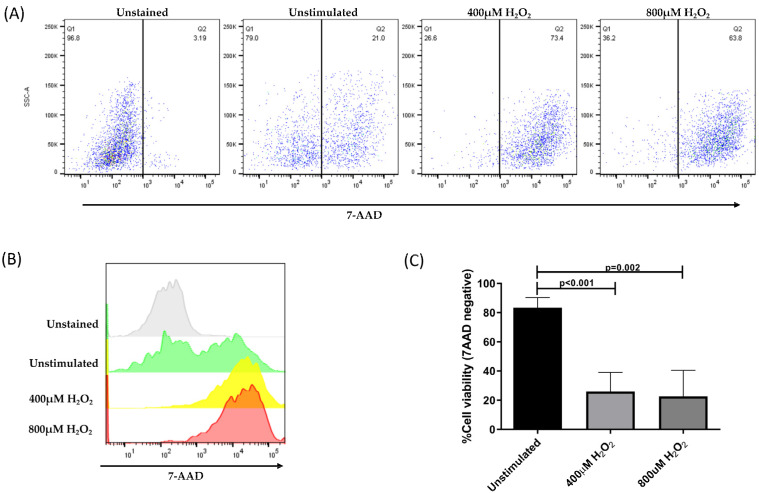
Percentage of viable ARPE-19 cells post-treatment, with different concentrations of H_2_O_2_ for 24 h to induce a state of oxidative stress. Viable cells exclude the 7-AAD dye; whereas, non-viable cells allow dye penetration and intercalation with DNA. (**A**) Representative dot plots of cells without staining, unstimulated and stimulated with 400 μM of H_2_O_2_ and 800 μM of H_2_O_2_. (**B**) Histogram representation of 7-AAD mean fluorescence intensity in all evaluated conditions. (**C**) Mean ± standard deviation (SD) for three independent viability assays. The ANOVA test was performed to compare statistical differences between groups.

**Figure 5 antioxidants-12-01540-f005:**
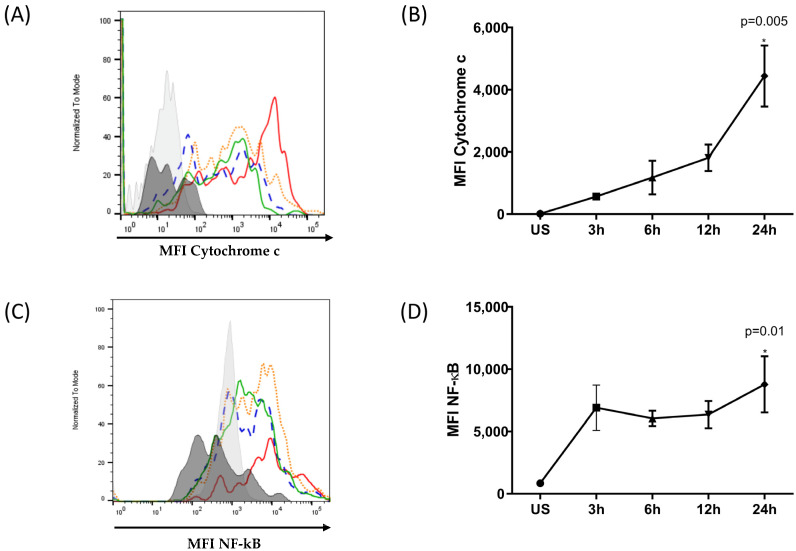
Kinetics of cytochrome C release and NF-κB expression in ARPE-19 cells under oxidative stress. ARPE-19 cells were stimulated with 400 μM of H_2_O_2_ for 3, 6, 12, and 24 h or were left unstimulated. Intracellular cytochrome C release and NF-κB expression were evaluated. (**A**,**C**) Representative histograms depicting MFI of cytochrome C and NF-κB, respectively. Black: unstained cells, grey: unstimulated cells, green: stimulated for 3 h, blue: stimulated for 6 h, orange: stimulated for 12 h, red: stimulated for 24 h. (**B**,**D**) The graphic depicts the mean MFI ± SD for three independent assays of cytochrome C and NF-κB, respectively. The ANOVA test was performed to compare statistical differences between groups. (* statistically significant).

**Figure 6 antioxidants-12-01540-f006:**
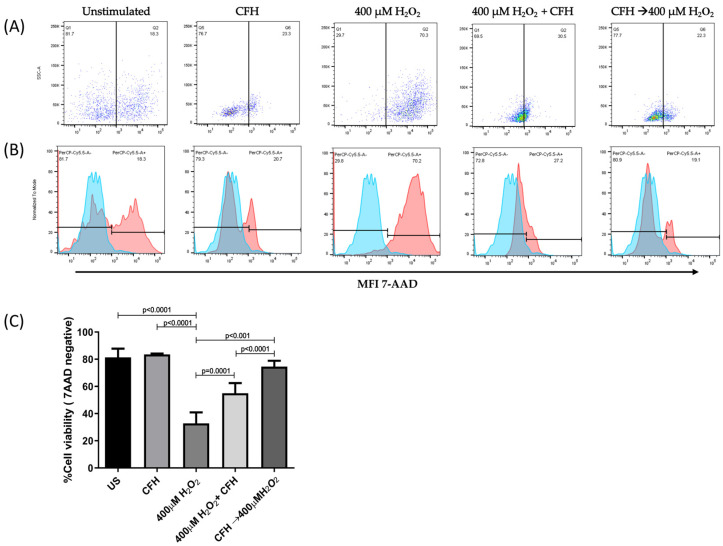
The percentage of viable ARPE-19 cells post-treatment with different stimuli. The ARPE-19 cells were cultured for 24 h in the presence of exogenous CFH, H_2_O_2,_ a combination of simultaneous H_2_O_2_ and CFH, or sequential CFH (1 h) followed by H_2_O_2_; they were stained with 7-AAD for analysis by flow cytometry. Results are representative of three independent assays. (**A**) The dot blot represents the percentage of viable and dead cells post-treatment. (**B**) The histograms represent the MFI of viable and dead cells. (**C**) The mean ± SD for three independent viability assays. The ANOVA test was performed to compare statistical differences between groups.

**Figure 7 antioxidants-12-01540-f007:**
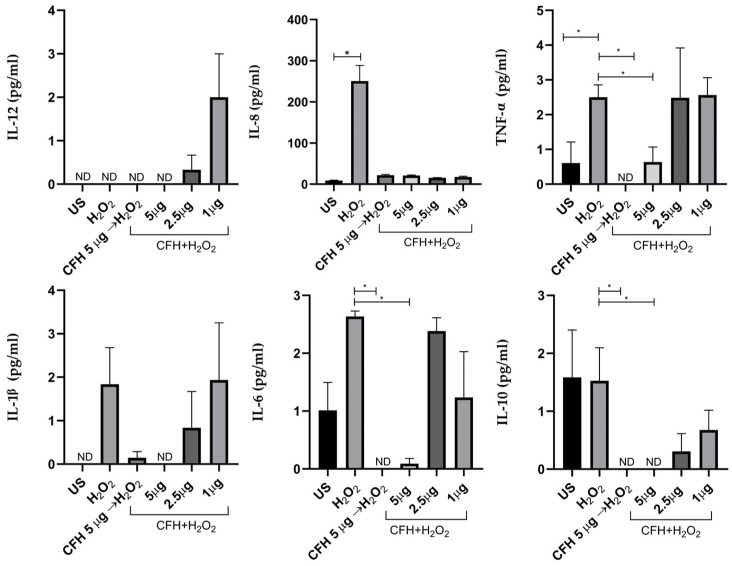
The profile of the pro- and anti-inflammatory cytokines in ARPE-19 cells secreted in response to oxidative stress and the addition of exogenous CFH. The cells were cultured and treated with different stimuli for 24 h and the cytokines in the culture supernatants were determined. The graphs are representative of three independent experiments. CFH was added to the culture 1 h before H_2_O_2_; CFH and H_2_O_2_ were added simultaneously. Simultaneous treatment of cells with different concentrations of CFH (1, 2.5, or 5 µg/mL) and 400 μM of H_2_O_2_. Unstimulated cells (US) were used as a control. The ANOVA test was performed to compare the differences between groups. * *p* < 0.05.

**Figure 8 antioxidants-12-01540-f008:**
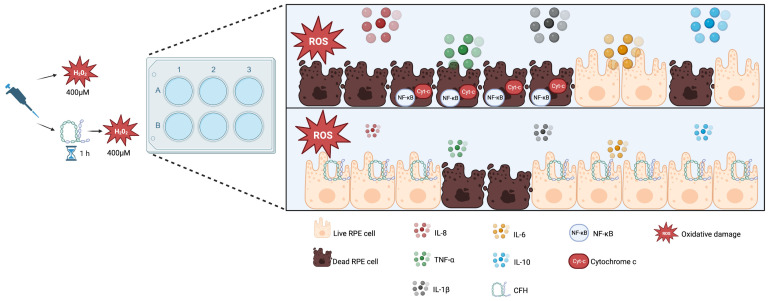
Model explaining the role of exogenous functional CFH in RPE under oxidative stress. RPE cells under oxidative stress increase NF-κB expression and secrete proinflammatory cytokines (IL-8, IL-6, IL-1β, TNF-α) and IL-10. Furthermore, cytochrome C is also released and detected in the cytoplasm of apoptotic cells. The addition of exogenous CFH protects cells from oxidative stress-related death and diminishes cytokine production. Created with BioRender.com.

## Data Availability

All data generated or analyzed during this study are included in this published article and its Appendix A.

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
