# Peer review of "Exogenous CFH Modulates Levels of Pro-Inflammatory Mediators to Prevent Oxidative Damage of Retinal Pigment Epithelial Cells with the At-Risk CFH Y402H Variant"

_antioxidants, 2023, doi:10.3390/antiox12081540_

Round 1

Reviewer 1 Report

Velazquez-Soto et al. present an original research paper titled "Exogenous CFH Modulates Levels of Pro-Inflammatory Mediators to Prevent Oxidative Damage of Retinal Pigment Epithelial Cells with the At-Risk CFH Y402H Variant." In this study, they investigated the role of CFH in the proinflammatory response using an in vitro model of oxidative stress in RPE cells with the at-risk CFH Y402H variant. While their efforts should be appreciated, the presentation of this research work is unsatisfactory. Upon reading, it raises the concern whether all the ARPE19 cells possess the At-Risk CFH Y402H Variant, as these cells are commonly used in various research studies under the assumption that they represent healthy human RPE cells. How the concentrations of CFH (1, 2.5, 5 ug/ml) were decided for the experiments should be explained. Furthermore, the meaning of "RTE" is unclear within the context of this study. The bands in the Western blot analysis are not clear, and supplementary figures should be included as one of the main figures. The results should only present data, without including parts of the methods or discussion. Overall, there are several issues that the authors should address to improve the presentation of their research work. The authors must clearly and sufficiently describe future works, limitations, and the clinical use of this research work. 

Need extensive editing to clarify every point.

Reviewer 2 Report

The widely well-written manuscript «Exogenous CFH Modulates Levels of Pro-Inflammatory Mediators to Prevent Oxidative Damage of Retinal Pigment Epithelial Cells with the At-Risk CFH Y402H Variant» by H. Velaquez-Soto and coworkers reports interestring in vitro data about the biological function of CFH. Technically, the set up is sound and the experiments have been well performed.

What makes interpretation of their results difficult is that according to their description of methods, they used the well-known and commercially available ARPE-19 cell line. What is new to me is that this well-established standard cell line for RPE cells culture is equipped with the at risk CFH Y402H variant, which is confirmed by the Sanger sequence. This needs to be clearly stated and supported by independent references since this is to my knowledge no genereally accepted knowledge. If they modified these cells to introduce the at-risk variant, this needs to be described in methods. If so, the data aree less useful until they are directly compared to ARPE-19 cells not harboring this variant.

Furthermore, the authors indicate that CFH secreted by these cells insufficiently may be less functional to compensate for oxidative stress, while exogenously added one does well. If they wish to prove this, they might consider to add an experiment directly comparing RPE cells without and with the at risk CFH Y402H variant in their response to oxidative stress. The wording in line 342-3 (CFH synthesized by ARPE-19 cells possessing the at-risk CFH Y402H variant may not be completely efficient) needs to be adopted accordinglx since endogenous CFH has not been quantified.

Further comments:

-      Most sub-chapters of chapter 3 are introduced by a small introduction. This may be helpful to better understand the results, but does not belong into this chapter lines 207-11, 238-43, 252-8,282-7,311-8, 353-8). Lines 232-5 should be moved tot he discussion.

-      That the induction of oxidative stress in the reported set up is not dose-dependent, may be linked to supraphysiological concentration levels chosen for H2O2 in the experiments. Response to supraphysiological oxidative stress may induce a response that possibly differs to that in a physiological range.

-      Figure 2B is difficult to interpret and seemingly not in line with the barr graph in 2C. Units in the x axis missing.

-      The reported cytokine concentrations widely range around the manufacturer-reported detection limit of ~1pg/ml of this assay. Has the number of wells in which single cytokines have not been detected been taken into account? Please add this information to allow the reader an esitmation oft he robustness of these results. Furthermore, the biological relevance of changes in theses cytokines (except maybe IL-8 the induction of which is widely suppressed by low doses of CFH though this finding is not supported by cytokine analyses in aqueous humor from humans with nAMD) has to be acknowledged and the text (lines 401-21) to be correpondingly re-worded.

-      This becomes namely evident with IL-12 which was not secretred in response to H2O2, but to low concentrations of CFH which further supports the need for applying physiological concentrations of H2O2 and CFH to RPE if physiological responses are addressed.

-      Another statistical example states IL-6 which is induced by H2O2, the induction seemingly normalized by low and suppressed by high doses of CFH, but not by an intermediate one. It seems at hand that this is more likely a random observation because of its concentration at the detection limit than a true finding. The results regarding for TNFa may also be a matter of discussion how to interpret.

-      Consequently, the application of supraphysiological concentrations of H2O2 could be a major disadvantage of the chosen study setup because it does not mimic physiological oxidative stress as stated in line 457. Please adopt and critically discuss the findings of the study on this background.

-      The impact of a heterozyghotous CFH locus has baerely been strived and might be discussed regarding ist relevance to control the oxidative stress-induced inflammatory response in RPE.

-      At several points, references are mis-leading or falsely cited, as examples refs 14, 16, 21 in the intro. Pls check all refere3nces for accuracy.

-      Lines 437-40 to be removed, repeat from intro.

Taken together, this manuscript reports despite the discussable supraphysiological concentrations of H2O2 and external CFH an interesting set of experiments which I enjoyed reading. It deserves further refinement namely in the interpretation oft he results in order to strengthen the findings. In a future study, as already envisioned by the authors, functional similarities differences between endogenously induced CFH in the mutant version and exogenous physiological CFH might provide further insighty.

Minor edits

Reviewer 3 Report

Velazquez-Soto et al. realized a very interesting article describing the “Exogenous CFH Modulates Levels of Pro-Inflammatory Mediators to Prevent Oxidative Damage of Retinal Pigment Epithelial Cells with the At-Risk CFH Y402H Variant”. I consider the manuscript very interesting but, at the same time, I suggest several revisions needed to improve the reliability and the completeness of the paper:

Introduction:

  1. A brief explanation of some key concepts and terms might be helpful for non-specialist readers. For example, terms like "Complement factor H (CFH)", "Age-related macular degeneration (AMD)" could be briefly explained when they first appear.
  2. It would be useful to provide some information on the prevalence or importance of AMD, so that readers understand why this study is necessary.
  3. Your introduction states that little is known about the impact of the CFH Y402H variant on AMD's pathophysiology. It would be helpful to expand on this statement to specify what is currently known and what knowledge gaps your study aims to fill.

Methods:

  1. More detail about the study design and procedure is necessary for readers to replicate your work. Specifically, provide more information about the cell culture conditions, how oxidative stress was induced, and the methods for measuring cytokine production and cell death.
  2. Include information about statistical analyses. For example, how were the data analyzed? What statistical tests were used?

Results:

  1. Make sure all of your results are presented in a clear, logical order that follows the methods section. It would be helpful to separate this section into subheadings that align with each of the methods used.
  2. Any figures or tables mentioned should be properly referenced and explained in the text. The information in the figure should be summarized in the text, not just referenced.
  3. Whenever you report a result, always provide the corresponding statistical test result to support it.

Discussion:

  1. The discussion could benefit from a more explicit comparison of your results with those from previous studies. Where do your results agree or disagree with the existing literature, and what might be the reasons for this?
  2. In addition to discussing the implications of your results, also discuss their limitations. For example, are there any limitations to the methods used or to the interpretation of the results?
  3. You should also clearly state the future directions of this research field based on your results.

Conclusion:

Consider adding a conclusion section to summarize the main findings and implications of your work, and to restate how your work fills gaps in the current knowledge.

General:

  1. Consistently use past tense when describing what was done in the study, and present tense when discussing the implications and future directions.
  2. Make sure that all the references are correctly formatted and that all the information (authors, year, title, journal, volume, pages) is included.
  3. Review the manuscript for typographical errors and to ensure all terminology is used consistently.
  4. Consider having a colleague or professional proofreader review the manuscript for clarity and grammar.

Furthermore, I suggest adding data related to recent bulk transcriptomics studies investigating the vascular alteration impact on several pathologies, like IRDs and MAV/CCM, which present a strong involvement of many pathways cited by the authors in the presented paper. 

Extensive editing of English language required.

Reviewer 4 Report

The current manuscript aims to report that exogenous CFH modulates levels of pro-inflammatory mediators to prevent oxidative damage of retinal pigment epithelial cells with the at-risk CFH Y402H variant. Overall, in my opinion, this study is significant in the area of new ocular antioxidants. Therefore, the manuscript may be suitable for publication to this journal if the authors’ improvements are deemed adequate.

Specific comments:

1.         The authors should carefully clarify the differences in the academic contribution points between the current manuscript and the earlier reports (please refer to the following papers: #1 DOI: 10.1038/s41598-019-50420-9 & #2 DOI: 10.3390/biomedicines9070763).

2.         As stated by the authors, a single nucleotide polymorphism predicting a Y402H replacement in the gene coding complement factor H (CFH) is an important risk factor for AMD. But, the audiences are unaware of the underlying reason of selecting complement factor H (rather than complement factor B or other complement factors). Furthermore, it is difficult to understand the motivation of using CFH Y402H variant for investigation given that its correlation with AMD risk remains unclear. Please further specify.

3.         To establish an in vitro model of cell stress induction, ARPE-19 cells were treated with 400 μM and 800 μM H2O2 for 24 h. However, from the obtained results, there was no significant difference in the cell death between these two groups. Why the authors adopted these two concentration conditions for experimental induction of cell death?

4.         Please clarify the scientific meaning of “unstained” and “unstimulated” groups? Whether these two groups without H2O2 treatment may have any differences for experiments?

5.         In order to examine the release of apoptotic and inflammatory markers induced by oxidative stress in ARPE-19 cells, the authors measured cytochrome C and NF-κB. But, they did not further specify the underlying reasons of checking these two specific markers. Please improve.

6.         As stated by the authors, it is known that one of the main generators of oxidative stress in the RPE cells is H2O2. However, this important statement was not supported any documented reference. In fact, one recent report has demonstrated the use of H2O2-induced ARPE-19 cell model for evaluation of therapeutic efficacy of ocular nanomedicine (please refer to DOI: 10.1021/acsnano.2c05824). The authors are highly recommended to consider the inclusion of this relevant paper in the reference list to enrich and update the article content.

7.         As mentioned above, the authors performed a comparative study on H2O2 concentration (400 μM vs 800 μM). However, they actually used 400 μM of H2O2 for subsequent studies. Why?

8.         Please must do a careful check of data presentation shown in Figure 3A since the panels of “Unstimulated” and “CFH ® 400 μM H2O2” seem to be the same.

9.         It is quite difficult for the readers to understand the results of Figure 4. Why the IL-12 level in the H2O2 group did not increase to an elevated extent? Please carefully check the data presentation again.

10.      Furthermore, the authors demonstrated the anti-inflammatory effects on H2O2-treated cell cultures by measuring different markers such as IL-12, IL-8, and TNF-α. However, all the makers did not exhibit similar trend of anti-inflammatory level. Please justify.

11.      Please carefully check the superscript and subscript errors throughout the manuscript.

Round 2

Reviewer 1 Report

This improved version of the paper can be considered suitable for publication.

Reviewer 2 Report

The manuscript has significantly gained. No open issues from my side

ok for me

Reviewer 4 Report

The revised version has adequately addressed most of the critiques raised by this reviewer and is now suitable for publication in "Antioxidants".